# Monitoring Tomato Leaf Disease through Convolutional Neural Networks

Antonio Guerrero-Ibañez [1] and Angelica Reyes-Muñoz [2,*]

1    Faculty of Telematics, University of Colima, Colima 28040, Mexico
2    Computer Architecture Department, Polytechnic University of Catalonia, 08860 Barcelona, Spain
*    Correspondence: angelica.reyes@upc.edu

**Abstract:** Agriculture plays an essential role in Mexico's economy. The agricultural sector has a 2.5% share of Mexico's gross domestic product. Specifically, tomatoes have become the country's most exported agricultural product. That is why there is an increasing need to improve crop yields. One of the elements that can considerably affect crop productivity is diseases caused by agents such as bacteria, fungi, and viruses. However, the process of disease identification can be costly and, in many cases, time-consuming. Deep learning techniques have begun to be applied in the process of plant disease identification with promising results. In this paper, we propose a model based on convolutional neural networks to identify and classify tomato leaf diseases using a public dataset and complementing it with other photographs taken in the fields of the country. To avoid overfitting, generative adversarial networks were used to generate samples with the same characteristics as the training data. The results show that the proposed model achieves a high performance in the process of detection and classification of diseases in tomato leaves: the accuracy achieved is greater than 99% in both the training dataset and the test dataset.

**Keywords:** convolutional neural networks; deep learning; disease classification; generative adversarial network; tomato leaf

## 1. Introduction

Tomato is one of the most common vegetables grown worldwide and is a high source of income for farmers. The 2020 statistical report of the Food and Agriculture Organization Corporate Statistical Database (FAOSTAT) indicates that world tomato production was 186.821 million tons [1]. In Mexico, the tomato is one of the main crops within the national production, being considered as a basic ingredient both in Mexican cuisine and in general in the cuisine of various parts of the world. According to a report published by Our World in Data in 2020, Mexico is among the top ten countries with the highest production of tomatoes, with a production of 4.1 million tons per year [2]. The Mexican Ministry of Agriculture, Livestock, Rural Development, Fishing and Food (MALRDFF) through the AgriFood and Fisheries Information Service presented the report on Mexico's AgriFood Trade Balance, indicating that tomato is the second most exported agricultural product, with avocado taking the first place. Besides this, tomato production in Mexico has an annual variation of 5.3% from 2011 to 2020 [3]. However, production is affected by different circumstances. The Food and Agriculture Organization (FAO) estimates that crop diseases are responsible for losses ranging from 20 to 40% of total production [4]. Various diseases of the tomato plant can affect the product in terms of quantity and quality, thus decreasing productivity. Diseases can be classified into two main groups [5]. The first group of diseases is related to infectious microorganisms including viruses, bacteria, and fungi. These types of diseases can spread rapidly from plant to plant in the field when environmental conditions are favorable. The second group of diseases is caused by non-infectious chemical or physical factors including adverse environmental factors, physiological or nutritional disorders and

herbicide injury. While it is true that non-infectious diseases cannot spread from plant to plant, diseases can spread if the entire plantation is exposed to the same adverse factor [6].

Some special conditions can cause plant diseases. Specifically, there is a conceptual model known as the disease triangle which describes the relationship between three essential factors: the environment, the host and the infectious agent. If any of these three factors is not present, then the triangle is incomplete, and therefore the disease does not occur. There are abiotic factors such as air flow, temperature, humidity, pH, and watering that can significantly affect the plant. The infectious agent is a kind of organism that attacks the plant such as fungi, bacteria, virus, among others. The host is the plant which is affected by a pathogen. When these factors occur simultaneously, disease is produced [7]. Generally, diseases are manifested by symptoms that affect the plant from the bottom up and many of these diseases have a rapid spread process after infection.

Figure 1 shows some of the most common diseases affecting tomato leaves including mosaic virus, yellow leaf curl virus, target spot, two-spotted spider mite, septoria leaf spot, leaf mold, late blight, early blight, and bacterial spot.

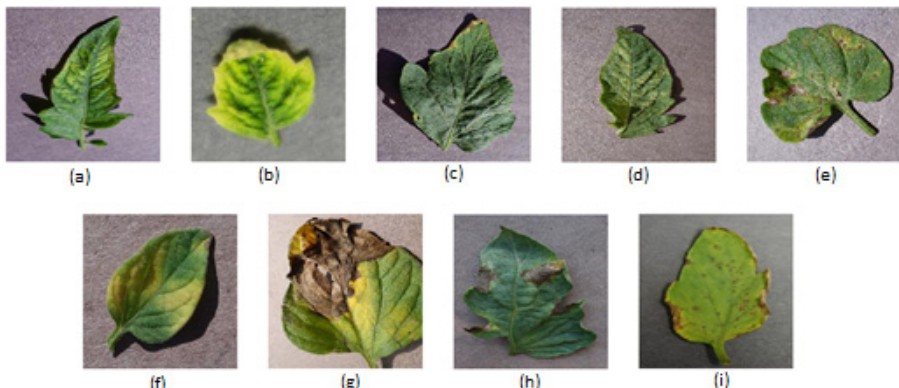

**Figure 1.** Representative images of the most common diseases affecting tomato leaves: (**a**) mosaic virus, (**b**) yellow leaf curl virus, (**c**) target spot, (**d**) two-spotted spider mite, (**e**) septoria leaf spot, (**f**) leaf mold, (**g**) late blight, (**h**) early blight and (**i**) bacterial spot.

Crops require continuous monitoring for early disease detection and thus the ability to apply proper mechanisms to prevent its spread and the loss of production [8].

The traditional methods used for the detection of plant diseases focus on the visual estimation of the disease by experts; studies of morphological characteristics to identify the pathogens; and molecular, serological, and microbiological diagnostic techniques [9]. The visual estimation method for plant disease identification is based on the analysis of characteristic disease symptoms (such as lesions, blight, galls and tumors) or visible signs of a pathogen (uredinospores of Pucciniales, mycelium or conidia of Erysiphales). Visual estimation is very subjective, as it is performed according to the experience of experts, so the accuracy of identification cannot be measured, and it is affected by temporal variation [10]. Microscopic methods focus on pathogen morphology for disease detection. However, these methods are expensive, time-consuming in the detection process and lead to low detection efficiency and poor reliability. In addition, farmers do not have the necessary knowledge to carry out the detection process, and agricultural experts cannot be in the field all the time to carry out proper monitoring.

New innovative techniques need to address the challenges and trends demanded by the new vision of agricultural production that requires higher accuracy levels and near real-time detection.

In recent years, different technologies such as image processing [11,12], pattern recognition [13,14] and computer vision [15,16] have rapidly developed and been applied to agriculture, specifically on automation of disease and pest detection processes. Traditional computer vision models face serious problems due to their complex preprocessing and design of image features that are time-consuming and labor-intensive. In addition, their

efficiency is conditioned by the accuracy in the design of feature extraction mechanisms and the learning algorithm [17].

Recently, the problem of plant disease detection has been addressed by deep learning technology, a subset of machine learning that is gaining momentum in disease identification due to the increase in computing power, storage capabilities and the availability of large data sets. Within the deep learning environment, one of the most widely used techniques for image classification, object detection and semantic segmentation are Convolutional Neural Networks (CNN) [18,19]. CNNs are useful for locating patterns in images, objects, and scenes by learning from the data obtained from the image for classification, eliminating the need for manual extraction of the features being searched for. CNN consist of several layers (such as convolutional, pooling and fully connected layers) to learn features from different training data [20,21]. This paper presents an architecture based on CNNs and data augmentation for early disease identification and classification in tomato leaves. The objective of the work is to implement a robust architecture that allows examining the relationship between the images of tomato leaves and the detection of a possible disease and performing a classification task to predict the type of disease with high accuracy levels.

The remainder of this article is organized as follows. Section 2 presents a brief discussion of previous research that has been conducted addressing the problem of disease identification in tomato. Section 3 explains in detail the CNN architecture proposed for tomato leaf disease identification and classification. A discussion of the experimental results obtained is presented in Section 4. Finally, Section 5 closes the paper with conclusions and future direction of the research work.

## 2. Related works

Plants disease detection has been studied for a long time. With respect to disease identification in tomatoes, much effort has been made using different tools such as classifiers focused on color [22,23], texture [24,25] or shape of tomato leaves [26]. Early efforts focused on support vector machines [27–30], decision trees [31,32] or neural network-based [33–35] classifiers. Visual spectrum images obtained from commercial cameras have been used for disease detection in tomato. The images obtained were processed under laboratory conditions, applying mechanisms such as stepwise multiple linear regression [36] and clustering process [37]. It is worth mentioning that the sample population for both works ranged between 22 and 47 for the first method and included 180 samples for the second experiment.

CNNs have rapidly become one of the preferred methods for disease detection in plants [38–40]. Some works have focused their efforts on identifying features with better quality through the process of eliminating the limitations generated by lighting conditions and uniformity in complex environment situations [41,42]. Some authors have developed real-time models to accelerate the process of disease detection in plants [43,44]. Other authors have created models that contribute to the early detection of plant diseases [45,46]. In [47], the authors make use of images of tomato leaves to discover different types of diseases. The authors apply artificial intelligence algorithms and CNN to perform a classification model to detect five types of diseases obtaining an accuracy of 96.55%. Some works evaluated the performance of deep neural network models applied to tomato leaf disease detection such as in [48], where the authors evaluated the LeNet, VGG16, ResNet and Xception models for the classification of nine types of diseases, determining that the VGG16 model is the one that obtained the best performance with an accuracy of 99.25%. In [49], the authors applied the AlexNet, GoogleNet and LeNet models to solve the same problem, obtaining accuracy results ranging between 94% and 95%. Agarwal et al. [50] developed their own CNN model based on the structure of VGG16 and compared it with different machine learning models (including random forest and decision trees) and deep learning models (VGG16, Inceptionv3 and MobileNet) to perform the classification of the 10 classes, obtaining an accuracy of 98.4%.

Several researches have focused on combining deep learning algorithms with machine learning algorithms to address and improve the accuracy of the classification problem,

for example, MobileNetv2 and NASNetMobile that were used to extract features from leaves and those features were combined with classification networks such as random forest, support vector machines and multinomial logistic regression [51]. Other works have applied algorithms such as YOLOv3 [45], Faster R-CNN [52,53] and Mask R-CNN [54,55] to detect disease states in plants.

Some efforts have been made to reduce the computational cost and model size such as Gabor filters [56] and K-nearest neighbors (KNN) [57] that have been implemented to reduce computational costs and overhead generated by deep learning. In [58], the authors reduced the computational cost by using the SqueezeNet architecture and minimizing the number of $3 \times 3$ filters.

## 3. Materials and Methods

In this section, we explain in detail the proposed architecture for the detection of diseases in tomato leaves. In general, the proposed architecture takes tomato leaves as input images and the output is a set of labels indicating (1) the type of disease in the image being analyzed or whether the leaf is healthy, (2) the label showing the predicted value obtained by our model, and (3) the prediction percentage.

Figure 2 shows the complete process of the algorithm that we applied for the process of detection and classification of diseases in tomato leaves. The global algorithm is composed of four stages: (a) creation of the experimental dataset, (b) creation of the proposed architecture, (c) distribution of the dataset, and (d) process of training and evaluation of the model.

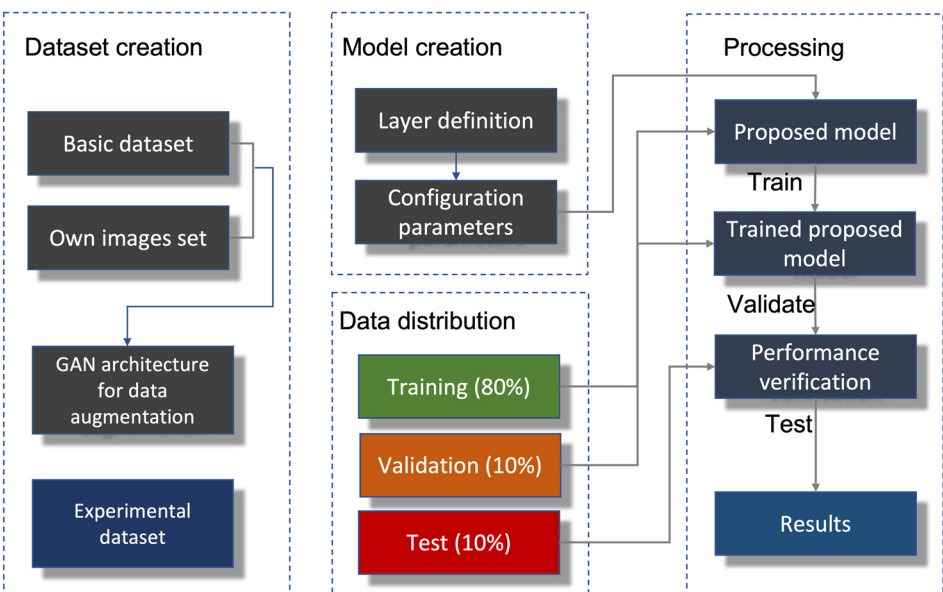

**Figure 2.** Representation of the proposed architecture for tomato disease detection.

### 3.1. Dataset Creation

As a first step, we proceeded to create the experimental dataset that would be used for training, validation, and performance evaluation of the proposed architecture. The public dataset available in [59] consists of 11,000 images that were the basis of our dataset. The images represent 10 categories, including nine types of diseases (tomato mosaic virus, target spot, bacterial spot, tomato yellow leaf curl virus, late blight, leaf mold, early blight, two-spotted spider mites, septoria leaf spot) and one category of healthy leaves. The dataset was complemented with 2500 images obtained from different crop fields in Mexico. The total number of images that made up our dataset was 13,500.

One of the problems that datasets face with deep neural network models is that when training the model, overfitting can occur, i.e., a model with high capacity may be able to "memorize" the dataset [60]. A technique known as data augmentation is used to avoid the

problem of overfitting. The goal of applying data augmentation is to increase the size of the dataset, and it is widely used in all fields [61]. Commonly, data augmentation is performed by two methods. The first method, known as the traditional method, aims to obtain a new image, which contains the same semantic information but does not have the ability of generalization. These methods include translation, rotation, flip, brightness adjustment, affine transformation, Gaussian noise, etc. The main drawbacks of these methods may be their poor quality and inadequate diversity.

Another method is the use of Generative Adversarial Networks (GANs), which are an approach to generative modeling using deep learning methods, such as CNNs, that aim to generate synthetic samples with the same characteristics as the given training distribution [62]. GAN models mainly consist of two parts, namely the generator and the discriminator [63]. The generator is a model used to generate new plausible examples from the problem domain. The discriminator is a model used to classify examples as real (from the domain) or fake (generated).

To create our experimental dataset, we made use of GAN to avoid the overfitting problem. To build our GAN, we define two separate networks: the generator network and the discriminator network. The first network receives a random noise, and from that number, the network generates images. The second network, the discriminator, defines whether the image it receives as input is "real" or not.

Because the images that complemented the dataset were not balanced for each category, the GAN network generated images that contributed to balance the dataset. The dataset was increased from 13,500 to 15,000 images, distributing the generated images in the different categories to create a balanced dataset.

### 3.2. Model Creation

Figure 3 shows the proposed CNN architecture for disease detection in tomato. The network has $112 \times 112$ color images as input, which are normalized to (0, 1) values. The proposed convolutional network has four convolutional layers that use filters whose values were 16, 32, 64, and 128, respectively. These values were assigned in that order since the layers closer to the beginning of the model learn convolutional filters less effectively than the layers closer to the result. In addition, the kernel size, which represents the width and height of the 2D convolution window, was set to a value of $3 \times 3$. This value was the recommended value for the number of filters to be used. Finally, rectified linear unit (ReLU) was used as the activation model for each convolved node.

After applying the convolutional layer, the maximum clustering layer was applied to down-sample the acquired feature map and condense the most relevant features into patches. This process is repeated for each of the convolutional layers defined in the architecture.

The result of the last MaxPooling layer is passed to a MaxAveragePooling layer to be converted to a column vector and connected to the dense layer of 10 output nodes (which represent the 10 categories) used as softmax activation. Each node represents the probability of each category for the evaluated image. Table 1 shows the information of the layer structure of the proposed model.



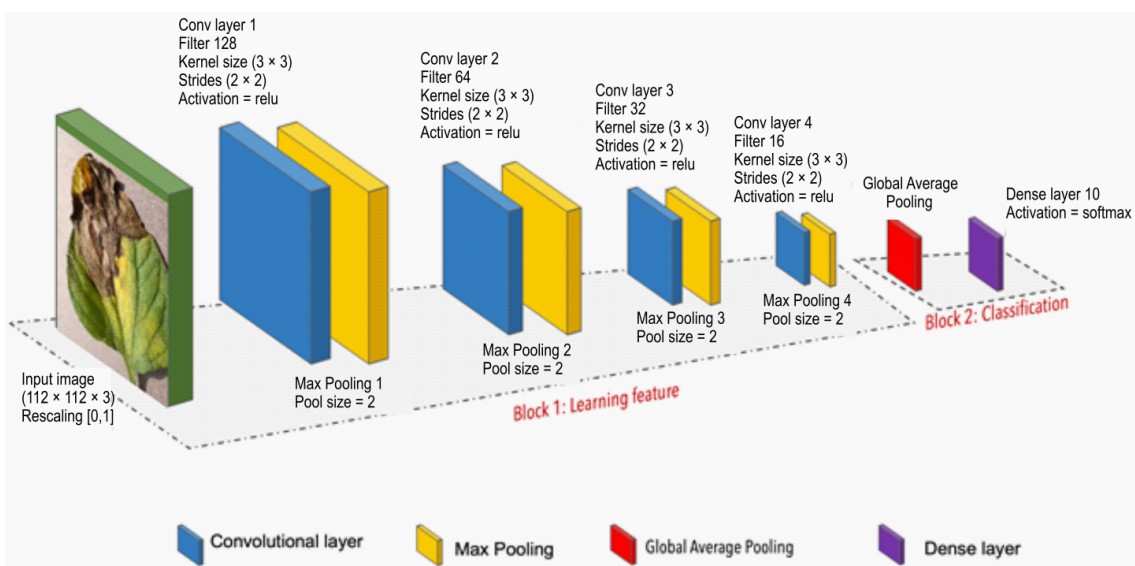

**Figure 3.** Representation of the proposed algorithm for tomato disease detection.

**Table 1.** Information on the layers structure of the proposed model.

| Layers | Parameters |
| --- | --- |
| Conv2D | Filters: 128, kernel size: (3,3), activation: "relu", input shape: (112,112,3) |
| MaxPool2D | Pool size: (2,2) |
| Conv2D | Filters: 64, kernel size: (3,3), activation: "relu" |
| MaxPool2D | Pool size: (2,2) |
| Conv2D | Filters: 32, kernel size: (3,3), activation: "relu" |
| MaxPool2D | Pool size: (2,2) |
| Conv2D | Filters: 16, kernel size: (3,3), activation: "relu" |
| MaxPool2D | Pool size: (2,2) |
| Dropout | Rate: 0.2 |
| GlobalAveragePooling2D | |
| Dense | Units: 10, activation: "softmax" |

### 3.3. Data Distribution

One of the most common strategies to split the dataset into training and validation sets is assigning percentages, for example, 70:30 or 80:20. However, one of the problems that can arise with this strategy is that it is uncertain whether high validation accuracy indicates a good model. When performing this division, it could happen that some information is missing in the data that are not used for training, causing a bias in the results.

We apply a k-fold cross-validation method to evaluate the performance of the model. The k-folds method tries to ensure that all features of the dataset are in the training and validation phases. The k-fold cross-validation method divides the dataset into subsets as k number. Therefore, it repeats the cross-validation method k times. Common values in machine learning are k = 3, k = 5, and k = 10. We use k = 5 to provide good trade-off of low computational cost and low bias in an estimate of model performance.

### 3.4. Model Creation

For the training process, we use Adam as the optimization algorithm. Adam updates network weights iterative based on training data. The loss function was categorical_crossentropy, one of the most used loss functions for multi-class classification models

where there are two or more output labels. The number of epochs for the training and validation process was 200. The steps_per_epoch parameter was 12,000, and for the validation the parameter it was 3000. Table 2 shows a summary of some of the parameters used for the training and validation phase.

**Table 2.** Training Parameters for the Proposed Model.

| Parameter | Value |
|---|---|
| Optimization algorithm | Adam |
| Loss function | Categorical cross entropy |
| Batch size | 32 |
| Number of epochs | 200 |
| Steps per epoch | 12,000 |
| Validation steps | 3000 |
| Activation function for conv layer | ReLu |

## 4. Results

In this section, we describe the scenario setup and the results obtained in the performance evaluation process of the proposed model.

### 4.1. Environmental Setup

Our model was developed in Google Collaboratory, a free Python development environment that runs in the cloud. Google Collaboratory is widely used for the development of machine learning and deep leaning projects. In our project, we use the following libraries: Tensorflow, an open-source library used for numerical computation and automated learning; Keras, a library used for the creation of neural networks; numpy, used for data analysis and mathematical calculations; matplotlib used for graph management and TensorBoard to visually inspect the different runs and graphs.

The model was trained with 200 epochs. We applied early stopping to monitor the performance of the model for the 200 epochs on a held-out validation set during the training to reduce overfitting and to improve the generalization of the neural network. For the evaluation of the model, the validation accuracy scheme allowed early stopping to be activated during the process.

Since our problem is a multi-class classification model, we use the Adam algorithm as the optimizing algorithm. In addition, the cross-entropy categorical loss function was used due to the nature of the multi-class classification environment. During the training process, we implemented checkpoints to save the model with the best validation accuracy, and thus be able to load it later to continue training from the saved state if necessary.

### 4.2. Evaluation Metrics

To analyze the performance of our model, the following four metrics were considered. The first metric to evaluate was accuracy, which represents the behavior of the model across all classes. Accuracy is calculated as the ratio between the number of correct predictions to the total number of predictions (Equation (1)).

Precision was our second metric, which represents the accuracy of the model in classifying a sample as positive. This parameter is calculated as the ratio of the number of positive samples correctly classified to the total number of samples classified as positive (Equation (2)).

We also analyzed the recall parameter, which measures the ability of the model to detect positive samples and is calculated as the ratio of the number of positive samples correctly classified to the total number of positive samples (Equation (3)).

Finally, we analyzed the $F1$ score parameter. This metric combines the precision and recall measures to obtain a single value. This value is calculated by taking the harmonic mean between precision and recall (Equation (4)).

The following equations were used to calculate accuracy, precision, recall and *F*1 score:

$$Accuracy = \frac{TP + TN}{TP + TN + FP + FN} \tag{1}$$

$$Precision = \frac{TP}{TP + FP} \tag{2}$$

$$Recall = \frac{TP}{TP + FN} \tag{3}$$

$$F1\ Score = 2 \times \frac{(precision \times recall)}{(precision + recall)}. \tag{4}$$

### *4.3. Results and Discussion*

In this section, we analyze the results obtained in the evaluation of the performance of the proposed CNN model in tomato crops. We compare our results with some of the proposed models published in the literature.

#### 4.3.1. Validation of the Proposed Model

The validation of the model was analyzed by applying the k-fold cross-validation procedure to estimate the performance of our algorithm on the tomato images dataset. We define a k-value of 5 to split the dataset. The Scikit-Learn machine learning library was used to implement the k-fold method returning a list of scores calculated for each of our five folds.

Figure 4 shows the results obtained by applying the k-folds method to evaluate the performance of the model. The results show a stability of the model, as it is observed in the different metrics analyzed. There is a very similar behavior with the five folds both in the training phase and in the validation phase, demonstrating that there is no overfitting in the proposed model.

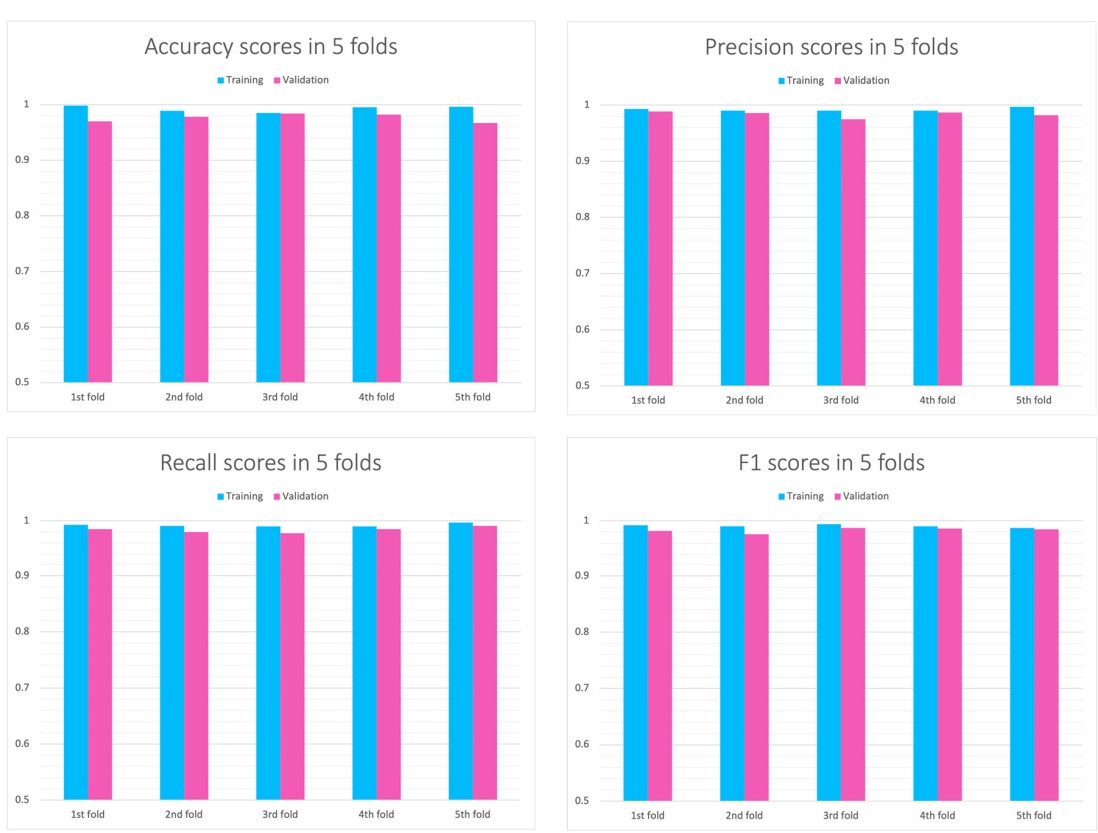

**Figure 4.** Results obtained for k-folds.

Figure 5 demonstrates the performance of our model in the training and validation stages for identification and classification of tomato leaf diseases. The results achieved a training accuracy of 99.99%. The time used for the training process was 6234 s in the MGPU (Multiple-Graphics Processing Unit) environment. The proposed model achieved a validation accuracy of 99.64% in leaf disease classification.

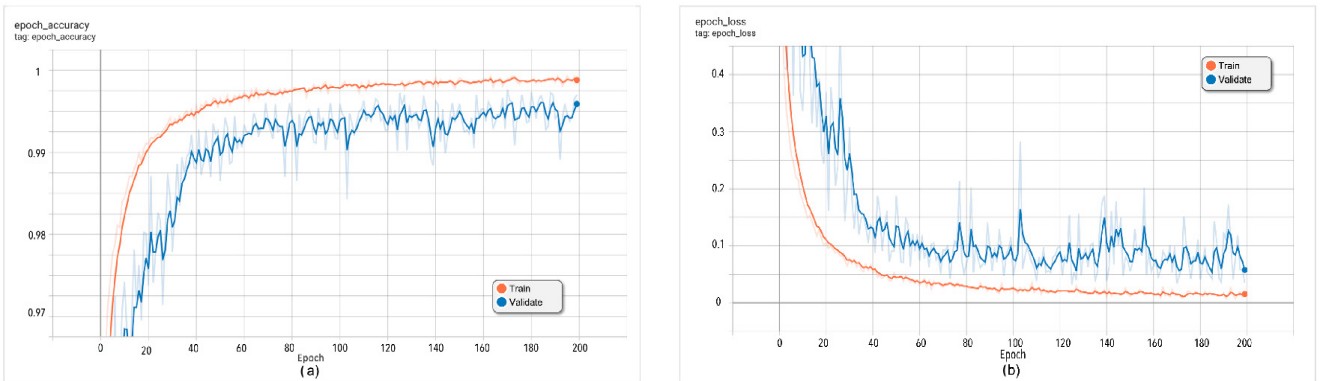

**Figure 5.** Results obtained of the proposed model during the training and validation phases: (**a**) accuracy and (**b**) loss.

Figure 6 shows the confusion matrix obtained in the evaluation of the proposed model. The confusion matrix shows the true positive (TP), true negative (TN), false positive (FP) and false negative (FN) values obtained for each class evaluated [64].

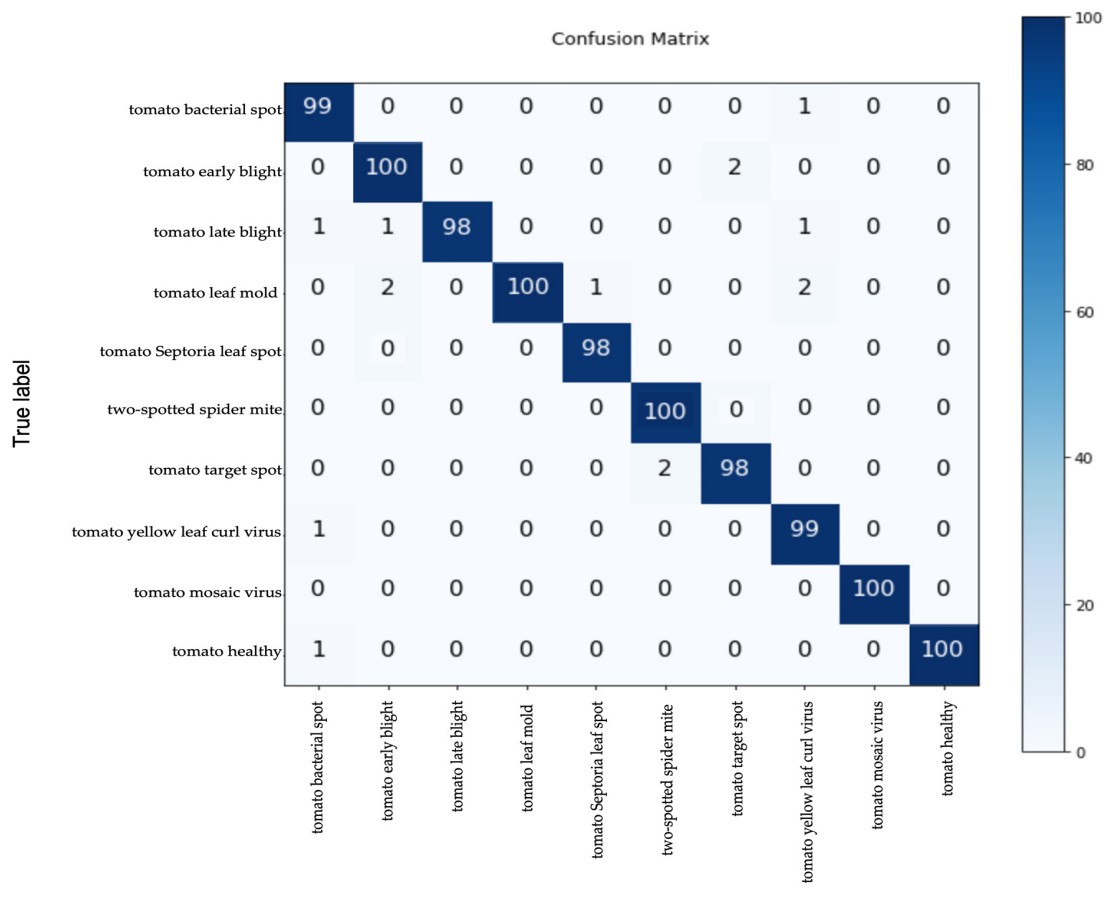

**Figure 6.** Confusion matrix of the proposed model.

According to the results, which are reflected in the confusion matrix, we can see that the proposed model was able to predict half of the classes that were evaluated using the test dataset with a 100% accuracy. For the rest of the classes, the model reached an accuracy level of at least 98%, thus obtaining better values than those of several of the works proposed in the literature.

Table 3 presents the results obtained in the classification performance of the proposed model on each of the classes defined within the experimental dataset. According to the data reflected in the table, the value obtained for the recall metric is high for each category defined in the dataset; this allows inferring the performance of the proposed model, which is able to correctly classify the corresponding disease with accuracy higher than 98%.

**Table 3.** Class-wise Performance of the Proposed Model.

| Class | Precision | Recall | F1 Score | Support |
|---|---|---|---|---|
| Tomato bacterial spot | 0.99 | 0.990 | 0.99 | 100 |
| Tomato early blight | 1 | 1 | 0.98 | 100 |
| Tomato late blight | 0.97 | 0.98 | 0.97 | 100 |
| Tomato leaf mold | 1 | 1 | 0.99 | 100 |
| Tomato Septoria leaf spot | 0.99 | 0.98 | 0.97 | 100 |
| Tomato Two-spotted spider mite | 1 | 1 | 0.98 | 100 |
| Tomato target spot | 0.99 | 0.98 | 0.99 | 100 |
| Tomato yellow leaf curl virus | 0.98 | 0.98 | 0.98 | 100 |
| Tomato mosaic virus | 1 | 1 | 0.99 | 100 |
| Tomato healthy | 1 | 1 | 0.99 | 100 |

The architecture and weights obtained from the proposed model were saved as a hierarchical data file to be used during the prediction process. The prediction process uses a dataset with a total of 1350 images. The matplotlib library was used to visualize the prediction result. For each prediction, the image, the true result, and the result of the prediction made with the proposed model were displayed, together with the percentage of accuracy. Figure 7 shows some results of the predictions made by the model.

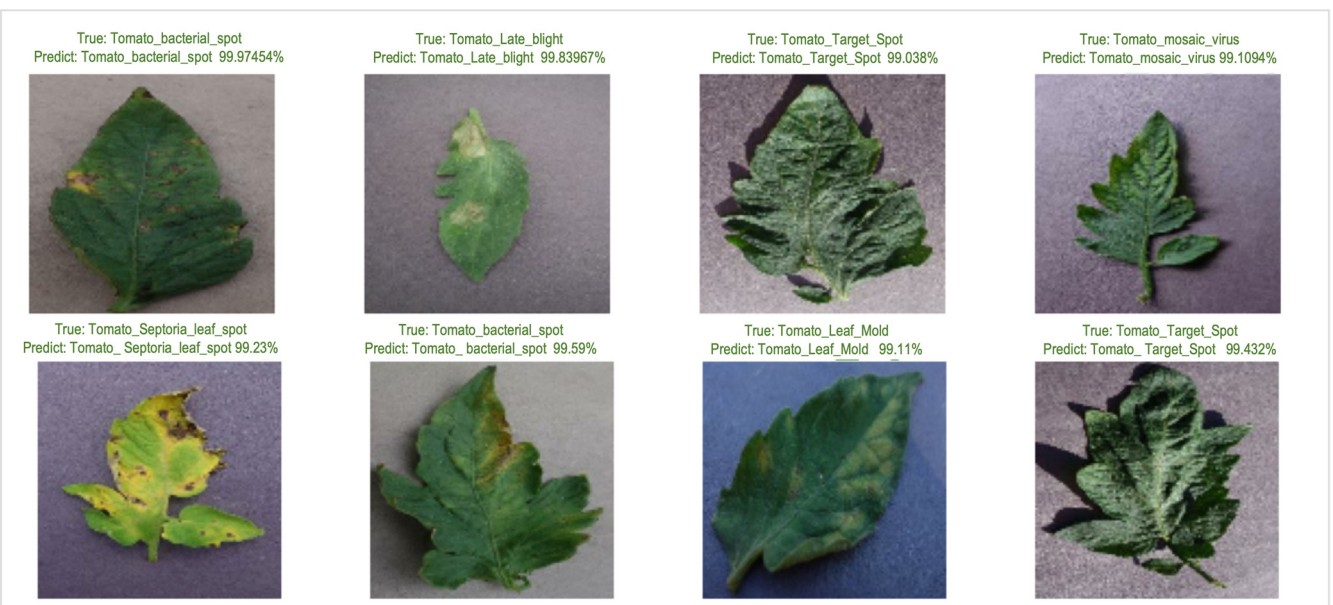

**Figure 7.** Sample predicted images using the proposed model.

4.3.2. Comparison of the Model

Finally, our model was compared with other techniques proposed in the literature (Widiyanto et al. [65], Afif Al Mamun et al. [66], Kaur et al. [67], AlexNet [68]; Inception-v3-

Net, ResNet-50 and VGG16Net [69]). Figure 8 presents the results of the comparison and shows that for the accuracy and recall metrics, the proposed model obtained the best results, reaching an accuracy of 99.9%. With respect to the precision metric, the proposed algorithm had a result only lower than the VGG16Net technique, but with a result of 0.99. For the F1 metric, the proposed model had a similar result to that of the VGG16Net technique.

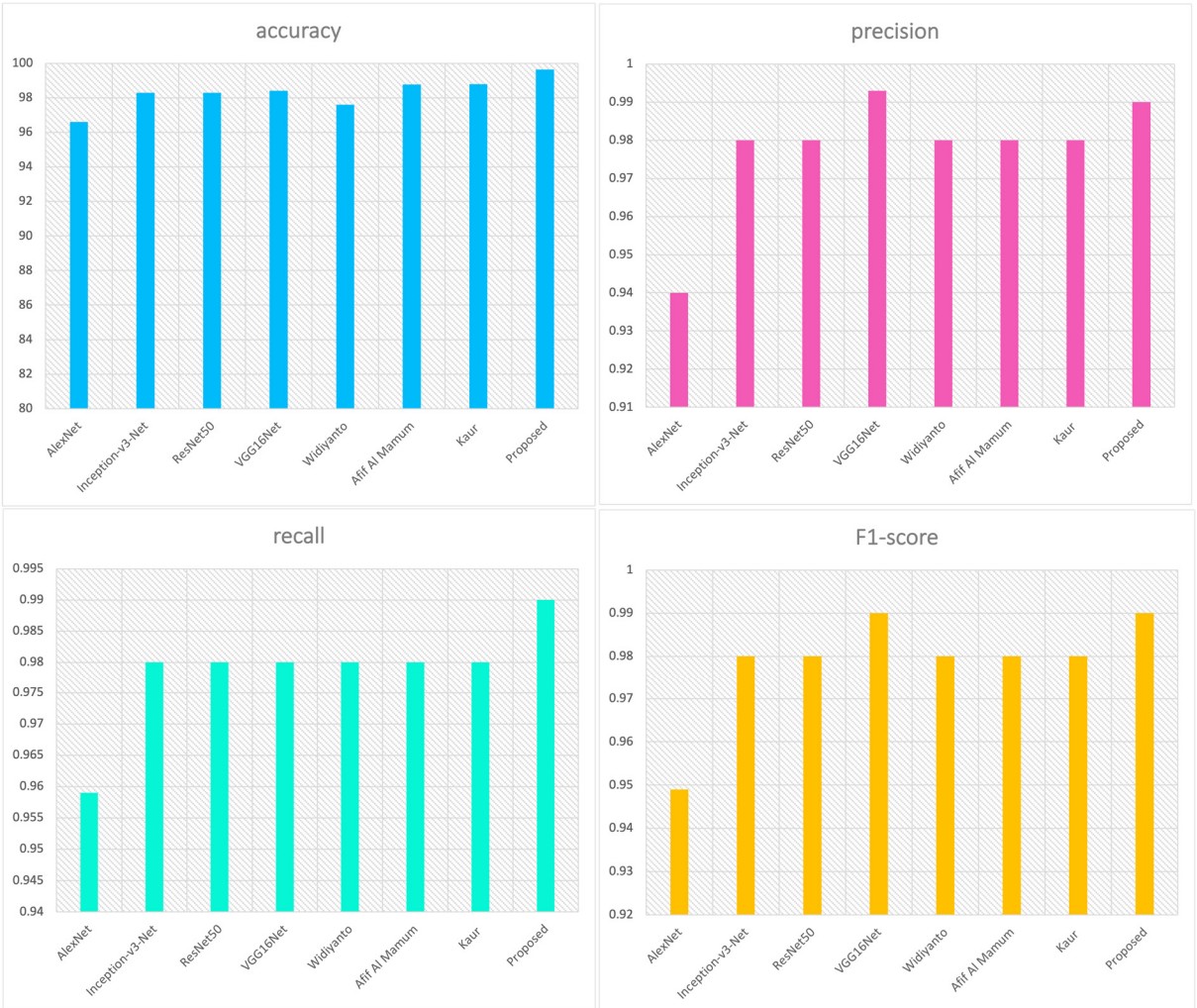

**Figure 8.** Performance comparison of proposed model and existing models.

In addition, a comparison was made of the complexity of the proposed model and some of the other models included in the comparison (data were not obtained for some of the models used in the comparison). Specifically, the number of trainable parameters and the size of the model were analyzed. The data obtained are shown in Table 4. Finally, Table 5 shows a summary of the performance of the models using the metrics accuracy, precision, recall and F1 score.

**Table 4.** Complexity comparison of proposed model and existing models.

|  | ResNet | VGG16Net | Inception-v3-Net | AlexNet | Proposed |
|---|---|---|---|---|---|
| Trainable parameters (Millions) | 26.7 | 39.4 | 24.9 | 44.7 | 5.6 |
| Model size (MB) | 98 | 128 | 92 | 133 | 36 |

**Table 5.** Class-wise Performance of the Proposed Model.

| Reference | Accuracy | Precision | Recall | F1 Score |
|---|---|---|---|---|
| Widiyanto, et al. (2019) | 97.6 | 0.98 | 0.98 | 0.98 |
| Afif Al Mamum et al. (2020) | 98.77 | 0.98 | 0.98 | 0.98 |
| Kaur et al. (2019) | 98.8 | 0.98 | 0.98 | 0.98 |
| Proposed model | 99.64 | 0.99 | 0.99 | 0.99 |

## 5. Conclusions

In this research, we propose an architecture based on CNNs to identify and classify nine different types of tomato leaf diseases. The complexity in detecting the type of disease lies in the fact that the leaves deteriorate in a similar way in most of the tomato diseases. It means that it is necessary to develop a deep image analysis to judge the types of tomato leave diseases with a proper accuracy level.

The CNN that we design is a high-performance deep learning network that allows us to have a complex image processing and feature extraction through four modules: the module dataset creation that makes an experimental dataset using public datasets and photographs taken in the fields of the country; model creation that is in charge of parameters configuration and layers definition; data distribution to train, validate and test data; and processing for the optimization and performance verification.

We evaluate the performance of our model via accuracy, precision, recall and the F1-score metrics. The results showed a training accuracy of 99.99% and a validation accuracy of 99.64% in the leaf disease classification. The model correctly classifies the corresponding disease with a precision of 0.99 and an F1 score of 0.99. The recall metric has a value of 0.99 on the classification of the nine tomato diseases that we analyzed.

The resulting confusion matrix describes that our classification model was able to predict half of the classes that were evaluated using the test dataset with a 100% accuracy. For the rest of the classes, the model reached an accuracy level of 98%, thus obtaining better values than those of several of the works proposed in the literature.

**Author Contributions:** Conceptualization, A.G.-I.; Methodology, A.G.-I. and A.R.-M.; Software A.G.-I.; Validation, A.G.-I. and A.R.-M.; Formal analysis, A.G.-I.; Resources, A.R.-M.; Data curation, A.G.-I.; Writing—review & editing, A.G.-I. and A.R.-M. All authors have read and agreed to the published version of the manuscript.

**Funding:** This work was partially funded by the State Research Agency of Spain under grant number PID2020-116377RB-C21.

**Data Availability Statement:** The datasets generated during the current study are available from authors on reasonable request.

**Conflicts of Interest:** The authors declare no conflict of interest.

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
