# Peer review of "Monitoring Tomato Leaf Disease through Convolutional Neural Networks"

_electronics, doi:10.3390/electronics12010229_

Round 1
Reviewer 1 Report
In this study, the authors proposed a 10-class classifier to classify 9 different diseases using photographs of tomato leaves.
According to my evaluations, there are some points I would like to draw attention to. These are the ones:
1. The authors designed a traditional sequential convolutional neural network. This structure does not contain enough innovation.
2. Comparisons of the proposed model with state-of-art models have not been made. This is an important shortcoming.
3. No k-fold cross validation was performed for the proposed model. In order to evaluate the performance of the model independent of the data set, k-fold cross validation should be performed.
4. The article mentions the use of GAN. However, it is said that 11000 of the 13500 images used were obtained using the publicly available dataset, and 2500 of them were obtained from tomato plants in Mexico. In this case, what is the place of the use of GAN in this study? This is not explained enough. New data is obtained with the use of GAN, but data from looking at the numbers are not included in this set. This situation should be explained.
5. There are many spelling/grammatical errors in the article. These need to be revisited. Some of these are, for example:
5a. The usage of "Hence, the need to improve crop yields."
5b. At many points in the article, "In-formation", "environ-mental", "bot-tom", etc. There is an incorrect use of the "-" symbol as in words. These must be corrected.
5c. In line 26, the long form of the acronym FAOSTAT should be given.
5d. The abbreviation SAGARPA is used for the words The Mexican Ministry of Agriculture and Rural Development in line 30. The correct abbreviation must be written.
5e. In line 102, the abbreviation CNN is given, but at many points in the rest of the article, the long form "Convolutional neural networks" words are used. Once the abbreviation is given, it must be used.
5f. The word "rotation" was duplicated on line 207.
6. Figure 4 shows the schematic obtained using a ready-made package, but this representation does not seem suitable for an academic study. Information on layers should be presented in a more appropriate format. "None" expressions in the figure are not appropriate.
7. Table 1, Table 2 and Table 3 are given as figures. It is more appropriate to add it in table format.
8. The use of (1), (2), (3) and (4) in lines 302, 306, 309 and 312 is not understood. If these are used as references to equations, this should be indicated.
9. Equation numbers should be right-justified when writing equations.
10. In line 331, the long form of the abbreviation MGPU should be given.
11. The label of Figure 5 is incorrect and should be checked.
12. It would be more understandable to write the names or abbreviations of the classes instead of numbers as labels in the confusion matrix given in Figure 6.
13. The metric definitions in lines 415 and 421 are not appropriate for the Conclusion section and should be removed.
Reviewer 2 Report
Authors presented a paper on using CNN to identify tomato leaf disease integrating public data and in-house data. The authors presented a CNN architecture to achieve classification and compared this method to several other methods. The paper is quite easy to follow but still needs proofreading as grammatic errors can be found here and there. There are several issues that authors need to address.
Firstly, authors compared the proposed method with 5 other methods. However, the results were not clearly presented nor explained. It appeared the authors compared the accuracy on different datasets and different classes. This makes the comparison not meaningful. The comparison should be made on the same data to do the same classification.
Secondly, authors need to present details or at least some ideas on computation speed of the method compared to other method. This can be an important factor that the method can be applied to real-time classification in field.
Thirdly, I assumed the proposed CNN method was implemented using existing libraries. Authors need to give more details on the implementation, including the packages used, development platform, etc. And it would be necessary to upload the implementation to GitHub.
Round 2
Reviewer 1 Report
I would like to thank the authors for taking into account my comments in the previous review. The publication seems to have improved slightly after the revision.
As minor errors, the following should be corrected:
There is no Figure 4 in the article, while there are 2 Figure 6.
There are 2 Table 3 in the article.
The Conclusion section should include brief information on the numerical data.
Author Response
Thank you for your valuable review. We address all of the suggested points
